

# No effects of the antiandrogens cyproterone acetate (CPA), flutamide and *p,p'*-DDE on early sexual differentiation but CPA-induced retardation of embryonic development in the domestic fowl (*Gallus gallus domesticus*)

Luzie Jessl[1,2] and  Jörg Oehlmann[1]

[1] Aquatic Ecotoxicology, Goethe University Frankfurt, Frankfurt am Main, Hesse, Germany
[2] R-Biopharm AG, Darmstadt, Hesse, Germany

## ABSTRACT

Because a wide range of environmental contaminants are known to cause endocrine disorders in humans and animals, *in vivo* tests are needed to identify such endocrine disrupting chemicals (EDCs) and to assess their biological effects. Despite the lack of a standardized guideline, the avian embryo has been shown to be a promising model system which responds sensitively to EDCs. After previous studies on the effects of estrogenic, antiestrogenic and androgenic substances, the present work focuses on the effects of *in ovo* exposure to *p,p'*-DDE, flutamide and cyproterone acetate (CPA) as antiandrogenic model compounds regarding gonadal sex differentiation and embryonic development of the domestic fowl (*Gallus gallus domesticus*). The substances were injected into the yolk of fertilized eggs on embryonic day one. On embryonic day 19 sex genotype and phenotype were determined, followed by gross morphological and histological examination of the gonads. Treatment with flutamide (0.5, 5, 50 µg/g egg), *p,p'*-DDE (0.5, 5, 50 µg/g egg) or CPA (0.2, 2, 20 µg/g egg) did not affect male or female gonad development, assessed by gonad surface area and cortex thickness in both sexes and by the percentage of seminiferous tubules in males as endpoints. This leads to the conclusion that antiandrogens do not affect sexual differentiation during embryonic development of *G. gallus domesticus*, reflecting that gonads are not target organs for androgens in birds. *In ovo* exposure to 2 and 20 µg CPA/g egg, however, resulted in significantly smaller embryos as displayed by shortened lengths of skull, ulna and tarsometatarsus. Although gonadal endpoints were not affected by antiandrogens, the embryo of *G. gallus domesticus* is shown to be a suitable test system for the identification of substance-related mortality and developmental delays.

Corresponding author
Luzie Jessl, jessl@bio.uni-frankfurt.de

## INTRODUCTION

Among the substances in constant use, there is a group of chemicals with structural similarity to natural sex hormones. Contaminants with hormonal action, so called endocrine disrupting chemicals (EDCs), are suspected to affect the development and health status of humans and animals with special focus on sex differentiation and reproduction. As agonists and antagonists of androgen (AR) and estrogen (ER) receptors, EDCs can activate or block corresponding receptors, potentially affecting all systems controlled by the endocrine system. A growing number of reports underlines the assumption, that EDCs pose a threat to the ecosystem and to animal and human health (*Delbes et al., 2022*; *Ho et al., 2022*; *Marlatt et al., 2022*; *Metcalfe et al., 2022*). In order to assess possible effects and to weigh risks, the testing of chemicals for their endocrine potential is of great importance.

Currently, there are several internationally standardized biotests for the testing of androgenic and estrogenic EDCs in mammals, among them two frequently used rodent-based tests, namely Hershberger assay (*OECD, 2009*) and uterotrophic assay (*OECD, 2007*). Since mainly juvenile and adult animals with full pain perception are used in these tests, the search for a suitable animal replacement system is of great significance. Moreover, these tests do not adequately reflect the impact of EDCs on the most sensitive stage of life, the developing embryo.

There is a long tradition of using avian embryos to study sexual development and potential effects of environmental pollutants including EDCs (*Berg et al., 1998*; *Berge et al., 2004*; *Biau et al., 2007*; *Eising et al., 2001*; *Estermann, Major & Smith, 2021*; *Fry & Toone, 1981*; *Ioannidis et al., 2021*). It is well known that the exposure of xenobiotics during avian embryonic development can induce irreversible deformities or malformations of the sex organs and disrupt sex-specific behavior (*Farhat et al., 2020*; *Guioli et al., 2020*; *Harnett et al., 2021*; *Mentor et al., 2020a*; *Mentor et al., 2020b*; *Ottinger et al., 2008*; *Quinn, Summitt & Ottinger, 2008*; *Sharin et al., 2021*). One advantage is that the avian egg can be considered as a closed system lacking any exchange with its environment except for the interchange of gases. The single administration of a specific and standardized dose, often injected directly into the egg (*Berg et al., 1999*), may be sufficient to affect the developing embryo (*Davies et al., 1997*; *Gooding et al., 2003*; *McAllister & Kime, 2003*; *Zhang et al., 2007*). Since no exchange or loss of the substance is possible, this injection results in chronic chemical exposure.

The embryo of domestic fowl (*Gallus gallus domesticus*) is particularly suitable for our experiments as its developmental stages are fully described (*Hamburger & Hamilton, 1992*; *Keibel & Abraham, 1900*; *Starck & Ricklefs, 1997*).

Since there is no standardized procedure available, the present study is part of a project aiming to expedite a protocol to assess the potential effect of EDCs on early sexual differentiation in the chicken embryo. In earlier publications we presented the effect of estrogens, antiestrogens, and androgens on embryonic gonad sex development (*Jessl, Scheider & Oehlmann, 2018*; *Jessl et al., 2018*; *Scheider et al., 2018*). In the present study we finally analyzed the effects of antiandrogenic compounds on embryonic development. Cyproterone acetate (CPA), flutamide and *p,p'*-dichlorodiphenyldichloroethylene (*p,p'*-

DDE) were chosen as antiandrogenic model compounds as they are well studied and have been shown to cause adverse effects in various non-target organisms, including the bird embryo.

An important class of antiandrogens are synthetic drugs such as CPA and flutamide that were specifically designed to competitively bind to androgen receptors (AR) *in vivo* (*Bhatia et al., 2014b*). These AR antagonists have been used to treat androgen-dependent prostate cancer in men and menstrual cycle irregularities in women with polycystic ovarian syndrome (*Heinlein & Chang, 2004*; *Paradisi et al., 2013*). Furthermore, both compounds have been used for the testing of potential effects on different non-target organisms including the bird embryo (*Adkinsregan & Garcia, 1986*; *de Gregorio et al., 2021*; *Fitzgerald et al., 2020*; *Gismondi et al., 2019*; *Jin et al., 2019*; *Mentessidou et al., 2021*; *Rangel, Sharp & Gutierrez, 2006*; *Rolon et al., 2019*; *Utsumi & Yoshimura, 2009*; *Utsumi & Yoshimura, 2011*; *Yu et al., 2020*; *Yu et al., 2021*).

*P,p'*- DDE is the primary metabolite of the insecticide DDT, which was used widely from the 1940s until its ban in most industrial countries in the 1970s (*Quinn, Summitt & Ottinger, 2008*). DDT and its metabolites are known to induce eggshell thinning and developmental disorders in fish-eating and raptorial seabirds (*Bouwman et al., 2019*; *Buck et al., 2020*; *Fry & Toone, 1981*; *Hickey & Anderson, 1968*; *Holm et al., 2006*; *Peakall & Lincer, 1996*; *Ratcliffe, 1970*) and affect sexual development in quail and chicken (*Blomqvist et al., 2006*; *Halldin et al., 2003*; *Kamata, Shiraishi & Nakamura, 2020*; *Quinn, Summitt & Ottinger, 2008*). Furthermore, DDT and its main metabolite *p,p'*- DDE can act as endocrine disruptors in humans and wildlife. Their endocrine effect is mainly mediated by agonistic or antagonistic interaction with nuclear receptors (*Burgos-Aceves et al., 2021*). Depending on the organism and endpoint studied, the existing literature indicates that *p,p'*- DDE may have various effects on the endocrine system. While some authors find evidence that *p,p'*-DDE has an antiandrogenic mode of action in mammals *in vitro* and *in vivo*, other studies report an estrogenic mode of action. In the study of *Hoffmann & Kloas (2016) p,p'*- DDE exhibits both a slight estrogenic and a marked antiandrogenic mode of action and alters the calling behavior of the amphibian *Xenopus laevis*. In human, there is evidence that *p,p'*-DDE has estrogenic effects on human breast tumors *in vivo* and increases the risk of breast cancer (*Muñoz-de Toro et al., 2006*).

As potent AR antagonists, CPA, flutamide and *p,p'*- DDE are potentially able to interfere with embryonic gonadal sex differentiation by binding to the AR and thereby preventing normal AR signaling. Several embryonic tissues of *G. gallus domesticus* are known to express the AR, among them brain, heart, liver, mesonephros, intestine, syrinx, cloaca, and gonads (*Gasc & Stumpf, 1981*; *Katoh, Ogino & Yamada, 2006*; *Tanaka, Izumi & Kuroiwa, 2017*). In the developing gonads of both sexes, AR mRNA levels gradually increase during embryonic development, with the highest AR expression detected at pre-hatching stage E21 (*Katoh, Ogino & Yamada, 2006*). It should also be investigated whether *p,p'*- DDE has antiandrogenic and/or estrogenic effects leading to disruption of gonadal sex differentiation of the embryo of *G. gallus domesticus*.

The present study is part of a project aiming to advance a replacement method for testing hormonally active compounds in birds, using fertilized eggs of the domestic

fowl (*G. gallus domesticus*). The focus of the present work was on the investigation of potential gross morphological and histological changes of the gonads as well as the effect on morphometric endpoints of antiandrogen-treated E19 embryos. We hypothesized that (1) the examined morphometric endpoints (mortality, malformations, body lengths) are suitable for the detection of toxic effects of selected antiandrogenic endocrine disruptors on embryonic development of *G. gallus domesticus*, (2) *in ovo* exposure of *G. gallus domesticus* to selected antiandrogenic compounds results in teratogenic effects on the gonads of E19 embryos, and (3) the examined gonadal endpoints (gonad surface area, cortex thickness and percentage of seminiferous tubules) are suitable for detecting the effects of selected antiandrogenic endocrine disruptors on gonadal sex differentiation of the embryo of *G. gallus domesticus.*

The objective of the entire project is to investigate the sensitivity of the chicken embryo to different classes of EDCs and ultimately to develop a standardized test system as an alternative to bioassays with mammals according to the 3R principles (Replace, Reduce, Refine; *Russell & Burch, 1959*).

## MATERIALS AND METHODS

### Dosing

The experimental work was conducted in accordance to the principles of laboratory animal care and with the guidelines set by the European Communities Council Directive (86/609/EEC) and the German Regulations for Animal Welfare.

Cyproterone acetate (CPA; CAS: 427-51-0; purity: ≥98%), flutamide (CAS: 13311-84-7; purity: ≥99%) and *p,p'*- dichlordiphenyldichlorethen (*p,p'*- DDE; CAS: 72-55-9; purity: ≥98%) were purchased from Sigma Aldrich Chemie GmbH (München, Germany). Fertilized eggs of white Leghorn (*G. gallus domesticus*) were obtained from a local breeder (LSL Rhein-Main Geflügelvermehrungsbetrieb, Dieburg, Germany). Eggs were incubated at $37.5 \pm 0.5$ °C and $60 \pm 10\%$ relative humidity and turned over eight times a day in a fully automated incubator (J. Hemel Brutgeräte, Verl, Germany). CPA (0.2, 2, 20 µg/g egg), flutamide (0.5, 5, 50 µg/g egg) and *p,p'*- DDE (0.5, 5, 50 µg/g egg) were dissolved in 15 µL (CPA) or 60 µL (*p,p'*- DDE and flutamide) of the solvent dimethyl sulfoxide (DMSO; CAS: 67-68-5; purity: 99.5%; Applichem, Darmstadt, Germany). Injection of test substances was performed 24 h after the start of incubation as previously described in *Jessl, Scheider & Oehlmann (2018)* and *Jessl et al. (2018)*. For this purpose, a small hole was drilled at the widest diameter of the egg, allowing the application of the test substance into the egg yolk *via* syringe (Hamilton microliter syringes and needles; ga22s/51 mm/pst2). The injection hole was sealed with agarose gel (3%, in phosphate buffered saline). To identify and remove unfertilized eggs or dead embryos, eggs were periodically checked by candling.

Since the selected antiandrogenic substances have rarely been tested *in ovo* on the embryo of *G. gallus domesticus*, it was necessary to extrapolate the effect data from experiments with other bird species (mainly Japanese quail) or even other taxa (mainly mammals). After a thorough literature study, suitable dosage ranges for exposure experiments with the selected antiandrogenic substances were identified. Doses were chosen to be below acute toxicity but high enough to induce potential endocrine effects.

**Table 1  Real-time PCR profile.**

| Step | Description |
| --- | --- |
| Initial Denaturation (1x) | 15 min, 95 °C |
| PCR cycling (40x) | |
| Denaturation | 30 s, 95 °C |
| Annealing | 30 s, 52 °C |
| Elongation | 40 s, 72 °C |
| Final extension (1x) | 5 min, 72 °C |
| Melt curve (1x) | 60–95 °C, heating rate: 0.2 °C/s |

## Dissection, tissue preparation and evaluation

Dissection was performed on day 19 of incubation. All embryos were examined for external deformations and malformations of inner organs with special focus on ovaries and testes. Gonad surface areas were analyzed by determining the entire visible surface of each single gonad with an image editing program (Fiji is just ImageJ, Open Source). Fixation and further processing of the gonads was performed as previously described in *Jessl, Scheider & Oehlmann (2018)* and *Jessl et al. (2018)*. Bouin's solution was used for fixation. After 24 h, the fixative was replaced with 80% ethanol. Ethanol was removed using saccharose solutions with ascending concentration (10, 20, and 30%, in phosphate buffered saline), Subsequent sectioning of the gonads was performed using a cryomicrotome (Microm HM 500 O, Thermo Fisher Scientific Germany, Bonn, Germany; Selected settings: temperature $= -23$ °C, section thickness $= 6$ µm), with the gonads embedded in Tissue-Tek® (Sakura Finetek Europe B.V., Alphen aan den Rijn, Netherlands) for enhanced stability. The resulting thin sections were stained using hematoxylin and eosin.

## Determination of sexual genotype

A tissue sample taken from the heart during dissection, was used for DNA isolation. Embryos that died during the incubation period were also sampled before they were removed. All embryos were typed for their ZZ or ZW sexual genotype, using the PCR-based method of *Fridolfsson & Ellegren (1999)* and a modified protocol. DNA-amplification was performed using qPCR and the primers 2550F "5´-GTT ACT GAT TCG TCT ACG AGA-3´" and 2718R "5´-ATT GAA ATG ATC CAG TGC TTG-3´". The two primers mark two CHD1 introns, one located on the Z chromosome (CHD1Z, 600 bp) and the other on the W chromosome (CHD1W, 450 bp). The qPCR-mediated amplification and the subsequent melt curve were performed using EvaGreen® dye (see Table 1). Genetic sex determination was performed by melting curve analysis, which revealed characteristic sex-specific bands (*Chen et al., 2012*). Males showed a single fragment at 84 °C (600-bp CHD1-Z). Females showed a fragment at 84 °C (600-bp CHD1-Z) and a fragment at 82 °C (450-bp CHD1-W).

## Determination of embryonic energy metabolism

In order to study the impact of CPA-treatment on embryonic energy metabolism, protein, glycogen, and lipid contents were determined using a tissue sample from the liver taken

**Table 2** Concentrations of the standard solutions for the photometric determination of protein, glycogen, and lipid contents extracted from liver samples.

| Standard No. | Protein content | | Glycogen content | Lipid content |
|---|---|---|---|---|
| | 0.1% BSA solution[a] (μL) | 2% Na$_2$SO$_4$ solution[b] (μL) | 0.1% glucose solution[c] (μL) | 0.1% rapeseed oil solution[d] (μL) |
| 1 | 0 | 50.0 | 0 | 0 |
| 2 | 12.5 | 37.5 | 25 | 50 |
| 3 | 25.0 | 25.0 | 50 | 100 |
| 4 | 37.5 | 12.5 | 100 | 200 |
| 5 | 50.0 | 0 | 150 | 400 |
| 6 | – | – | 200 | – |

**Notes.**
[a] 100 mg BSA dissolved in 100 mL 2% Na$_2$SO$_4$ solution.
[b] 2 g Na$_2$SO$_4$ dissolved in 100 mL deionized water.
[c] 100 mg glucose dissolved in 100 mL deionized water.
[d] 100 mL rapeseed oil mixed with 100 mL chloroform.

during dissection. Six embryos of untreated and solvent control group as well as CPA-treated groups were used. For each fraction, calibration curves using standards in five (protein, lipid) or six (glycogen) ascending concentrations were established using 0.1% Bovine Serum Albumin (BSA) solution (protein), 0.1% glucose solution (glycogen) or 0.1% rapeseed oil solution (lipid) (see Table 2). All standard solutions were freshly prepared immediately before the start of extraction. Standards were treated analogously to samples. The liver was weighted and homogenized with a 2% sodium sulfate solution. Part of the homogenate was used to determine the protein content according to *Bradford (1976)*. For this purpose, 50 μL of the homogenate were transferred to a new tube. Samples and standards were mixed with 1.5 mL of Bradford reagent and then carefully inverted. Solutions were incubated for 5 min at room temperature and then transferred to a cuvette followed by photometric measurement. Another part of the homogenate was used to determine glycogen and lipid contents according to *Van Handel (1965)*, *Van Handel (1985a)* and *Van Handel (1985b)*. For this purpose, 100 μL of the homogenate were transferred to a centrifuge tube, mixed with 1.6 mL of a 1:1 mixture of chloroform/methanol and centrifuged for 2 min at 3,000 rpm. The resulting pellet, which was formed at the bottom of the centrifuge tube, contained the glycogen fraction, the supernatant contained the lipids. The supernatant was completely transferred to a new centrifuge tube, mixed with 0.6 mL deionized water, and centrifuged again for 2 min at 3,000 rpm. The resulting upper phase was discarded. The lower phase, which contained the lipid fraction, was set aside for later processing. Glycogen extraction was performed using the pellet from the previous mentioned step. For this, five mL of Anthron reagent (composed of 150 mL deionized water, 385 mL sulfuric acid, and 750 mg Anthron) were added to each pellet-containing centrifuge tube and standard. All tubes were sealed and incubated for 17 min at 95 °C in a water bath. The solutions were allowed to cool down and then transferred to a cuvette followed by photometric measurement. Lipid extraction was performed using the lipid-containing solutions previously set aside. To remove remaining solvent residues,

solutions were incubated at 95 °C with lids open in a water bath. Next, 200 µL of $H_2SO_4$ were added to each sample tube and standard. All tubes were sealed and incubated for 10 min at 95 °C in a water bath. Each tube received five mL of vanillin reagent (composed of 100 mL deionized water, 600 mL vanillin, and 400 mL $H_3PO_4$) followed by careful inversion. Finally, solutions were transferred to a cuvette and left to rest for five minutes, followed by photometric measurement.

The absorbance of standards and samples was determined using a photometer (BioSpectrometer®; Eppendorf, Hamburg, Germany; protein: 595 nm; glycogen/lipid: 625 nm). The absorbance of the respective standards was plotted against protein, glucose, or lipid content, respectively, calculating linear calibration curves. Based on the respective calibration curve, protein, glycogen, or lipid contents of the samples (µg/g liver) were determined and extrapolated to the total volume of the homogenate. Using the specific calorific values (protein: 17 kJ/g; glucose: 17 kJ/g; lipid: 37 kJ/g) the energy content of protein, glycogen, and lipid reserves in J/mg embryo was calculated.

## Measurements and statistics

Histological examination of embryonic gonads was performed as previously described in *Jessl, Scheider & Oehlmann (2018)* and *Jessl et al. (2018)*. For this purpose, a light microscope (Olympus BX50; Olympus, Tokyo, Japan) and a camera (JVC Digital Camera, KY-F75U, Yokohama, Japan) were used. Only left gonads were measured, and the cortex thickness of both sexes as well as the percentage of seminiferous tubules of males were determined. Thin sections originating from the gonad's middle sectional plate were evaluated ($n = 10$ per embryo). Five measurements per section were performed to determine the cortex thickness. Since cortex thickness is not constant over the whole organ, different representative areas around the gonad were chosen. The area of all seminiferous tubules in a defined image section was measured to determine a representative percentage of seminiferous tubules in the male left testis. For this a random representative image section was selected which showed only the medullary tissue but not the cortex region.

One experiment was performed with different concentrations of CPA (0.2, 2, 20 µg/g egg), another experiment was performed with different concentrations of flutamide and *p,p'*- DDE (0.5, 5, 50 µg/g egg). Solvent control was used as the reference-control. Data were analyzed using Fisher's exact test and one-way ANOVA with Dunnett's multiple comparison test with GraphPad Prism 5.03 (GraphPad Software Inc., San Diego, USA).

# RESULTS

## Embryonic mortality and malformations

The endpoint mortality provides an important indication of the toxicity of a substance, *i.e.,* the ability to produce adverse effects or to impair functions, which may ultimately result in the death of the organism under investigation. Toxicity depends not only on the substance itself, but also, among other factors, on the dosage used, the time of exposure and the duration of exposure.

*In ovo* exposure to all concentrations of *p,p'*- DDE and flutamide caused a concentration-dependent increase in embryonic mortality which was found to be significantly different

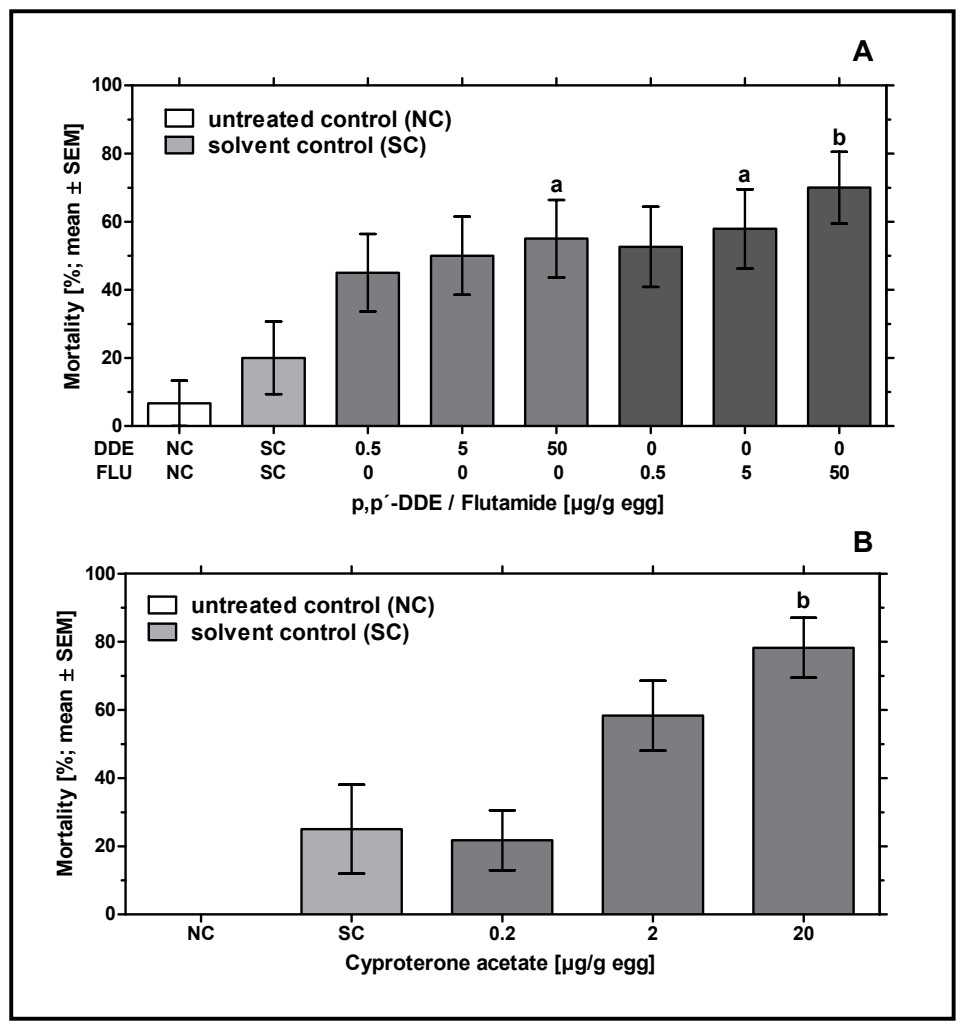

**Figure 1** Effects of *in ovo* exposure to *p,p'*- DDE and flutamide (A) or cyproterone acetate (B) on embryonic mortality of the domestic fowl (*Gallus gallus domesticus*) on embryonic day 19. Administered doses: *p,p'*- DDE (DDE): 0.5, 5, 50 µg/g egg; flutamide (FLU): 0.5, 5, 50 µg/g egg; cyproterone acetate (CPA): 0.2, 2, 20 µg/g egg. Statistical analysis by Fisher's exact test. NC: untreated control group. Lowercase indicates significant differences compared to the solvent control (SC). Level of significance: a, $p < 0.05$; b, $p < 0.01$.

from the solvent control for 50 µg *p,p'*- DDE/g egg ($p < 0.05$) and 5 and 50 µg flutamide/g egg ($p < 0.05$ and $p < 0.01$, respectively). Mortality rates of *p,p'*- DDE-treated groups were between 45% and 55%, mortality rates of flutamide-treated groups were between 53% and 70% (see Fig. 1A).

*In ovo* exposure to all concentrations of CPA resulted in a concentration-dependent increase in mortality which was found to be significantly different from the solvent control group for 20 µg CPA/g egg ($p < 0.01$). In the group treated with the highest concentration of CPA (20 µg/g egg), this resulted in nearly 78% mortality leaving only a few embryos for follow-up analyses (see Fig. 1B).

The endpoint *malformations* provide information on the teratogenicity of a substance, *i.e.,* the ability to cause irreversible malformations including growth retardation or death during the embryonic development of an organism. In this context, the endpoint *body length* can provide further indications as to whether a substance can cause growth retardation of individual body parts or even of the entire organism.

Different types of single or multiple malformations were found in the control, *p,p'*- DDE or flutamide-treated groups. In the untreated control group one embryo (6.67%) showed celosomia. In the solvent control two embryos (13.3%) showed either malformations of the extremities or celosomia. *In ovo* exposure to 0.5 µg *p,p'*- DDE/g egg led to two malformed embryos (10.5%), one of them with both-sided anophthalmia, the other one a "twin embryo" conjoined at the head, showing various malformations. *In ovo* exposure to 5 µg *p,p'*- DDE/g egg led to one malformed embryo (5.3%) with crossed beak and a cyclops-like eye at the front of the head. *In ovo* exposure to 0.5 µg flutamide/g egg led to two malformed embryos (10.0%) with celosomia, crossed beak and left-sided or both-sided anophthalmia while *in ovo* exposure to 5 µg flutamide/g egg led to one embryo (5.0%) with celosomia. In the highest concentration (50 µg/g egg) of both *p,p'*- DDE and flutamide, no malformations were detected. None of the *p,p'*- DDE- or flutamide-treated groups showed a statistically significant difference from the solvent or the untreated control group.

Different types of single or multiple malformations were found in the solvent control and CPA-treated groups. In the untreated control group, none of the embryos showed malformations. In the solvent control one embryo (8.3%) showed celosomia. *In ovo* exposure to 0.2 µg CPA/g egg led to one embryo (4.3%) with left-sided anophthalmia and *in ovo* exposure to 2 µg CPA/g egg led to two malformed embryos (8.3%) with exencephalia, right-sided anophthalmia and celosomia. None of the CPA-treated groups showed a statistically significant difference from the solvent or the untreated control group.

Remarkably, an increased incidence of significantly delayed development was found, which especially occurred at concentrations of 2 and 20 µg CPA/g egg. In order to analyze this statistically, the parameters length of skull (from the tip of the beak to the back of the head), length of ulna (right side) and length of tarsometatarsus (right side) were measured. Embryos exposed to the lowest concentration of 0.2 µg CPA/g egg showed no effects on body lengths, while higher concentrations of 2 and 20 µg CPA/g egg resulted in a statistically significant reduction of all three parameters ($p < 0.001$, respectively) compared to the solvent control (see Fig. 2A and Table 3).

To investigate the potential impact of CPA-treatment on embryonic energy reserves, we determined the content of protein, glycogen, and lipid in liver samples of control and CPA-treated groups. Compared to the solvent control, all CPA treatments were characterized by a significantly decreased content of glycogen (0.2 µg CPA/g egg: $p < 0.01$; 2 and 20 µg CPA/g egg: $p < 0.001$, respectively). In addition, embryos exposed to the highest concentration of 20 µg CPA/g egg showed a significantly decreased content of protein ($p < 0.001$). However, the content of lipids in this group was marginally but not statistically significantly increased ($p > 0.05$) (see Fig. 2B).

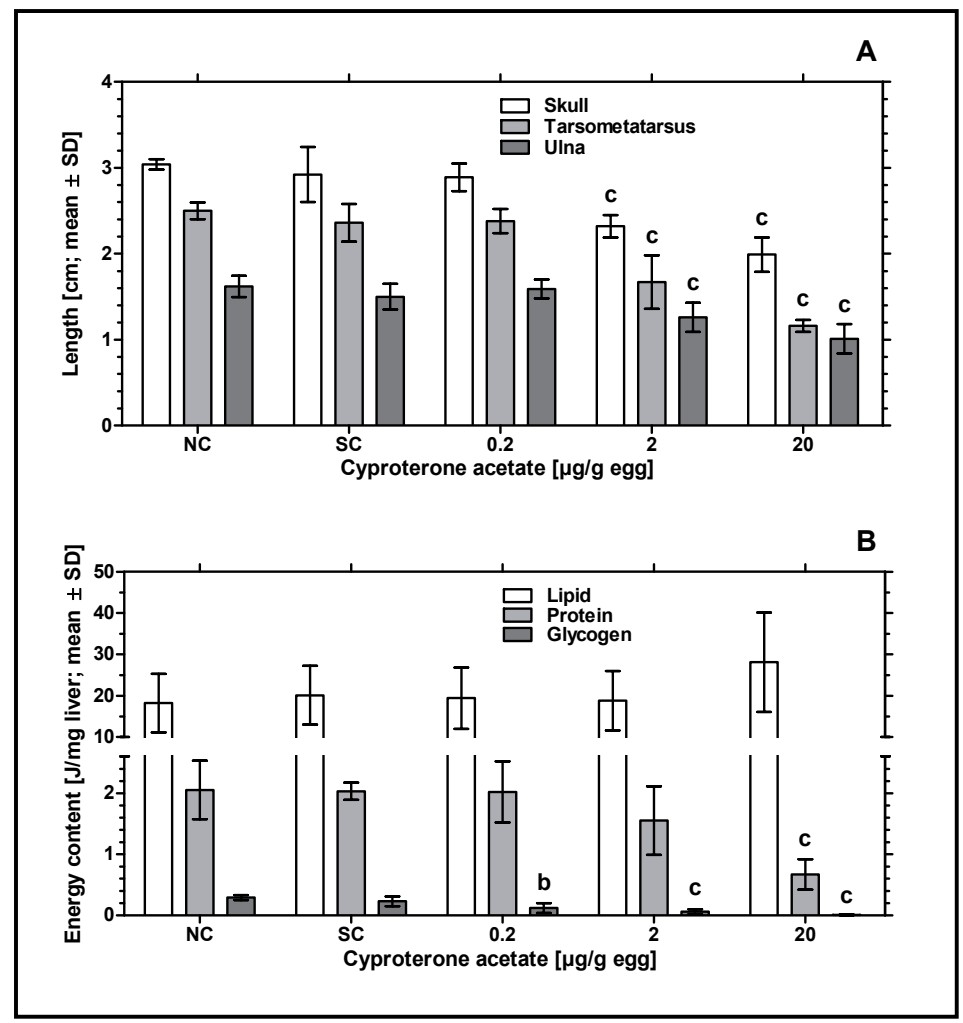

**Figure 2** **Effects of *in ovo* exposure to cyproterone acetate on body lengths (A) and energy levels (B) of embryos of the domestic fowl (*Gallus gallus domesticus*) on embryonic day 19.** Administered doses: cyproterone acetate (CPA): 0.2, 2, 20 μg/g egg. Endpoints shown: length of skull, tarsometatarsus and ulna (A) and energy levels (lipid, protein, and glycogen) of liver (B). Statistical analysis by one-way ANOVA with Dunnett's multiple comparisons test. NC: untreated control group. Lowercase indicates significant differences compared to the solvent control (SC). Level of significance: b, $p < 0.01$; c, $p < 0.001$.

## Morphological observation of the gonads—gonad surface area

Exposure to higher concentrations of *p,p'*- DDE or all concentrations of flutamide had no statistically significant effect on the surface areas of male or female gonads. Only *in ovo* exposure to 0.5 μg *p,p'*- DDE/g egg resulted in a statistically significant decrease of the surface area of the right ovary and a statistically significant increase of the surface area of the left testis ($p < 0.05$, respectively) compared to the solvent control. In control groups there was a statistically significant difference between the untreated and the solvent control for the female left gonad surface area ($p < 0.001$) and left and right male gonad surface areas

**Table 3** Lengths of skull, tarsometatarsus and ulna of E19 embryos of *Gallus gallus domesticus* after *in ovo* exposure to cyproterone acetate (CPA; 0.2, 2 or 20 $\mu$g/g egg).

| Substance ($\mu$g/g egg) | n | Length (cm; mean ± SD) | | |
| --- | --- | --- | --- | --- |
| | | Skull | Tarsometatarsus | Ulna |
| NC | 11 | 3.04 ± 0.06 | 2.50 ± 0.096 | 1.62 ± 0.122 |
| SC | 9 | 2.92 ± 0.32 | 2.36 ± 0.218 | 1.50 ± 0.149 |
| CPA 0.2 | 18 | 2.89 ± 0.16 | 2.38 ± 0.140 | 1.59 ± 0.110 |
| CPA 2 | 12 | 2.32 ± 0.13[c] | 1.67 ± 0.310[c] | 1.26 ± 0.170[c] |
| CPA 20 | 4 | 1.99 ± 0.20[c] | 1.16 ± 0.070[c] | 1.01 ± 0.170[c] |

Notes.

NC, untreated control group; SC, solvent treated control group.

Statistical analysis by one-way ANOVA with Dunnett's multiple comparisons test: Identical superscripted letters indicate significant differences compared to SC. Level of significance: c, $p < 0.001$.

($p < 0.05$, respectively) (see Fig. 3A). Compared to the solvent control, *in ovo* exposure to CPA did not affect male or female gonad surface areas (see Fig. 3B).

## Histological observation of the gonads—left testis and ovary

None of the examined antiandrogenic substances induced any effect on male or female gonadal sex differentiation. Neither *p,p'*- DDE, flutamide or CPA had a statistically significant effect on the percentage of seminiferous tubules in left testes or the cortex thickness in left testes or ovaries at any of the tested concentrations (see Fig. 4). The mean values for these endpoints in the antiandrogen-treated groups varied around the mean value of the respective solvent control group. The high mortality rate in the highest concentration of CPA (20 $\mu$g/g egg) resulted in a lack of usable tissue samples for the investigation of the histological parameters mentioned.

## DISCUSSION

### Embryonic mortality and malformations

*In ovo* exposure to higher concentrations of *p,p'*- DDE, flutamide and CPA resulted in significantly increased mortality rates. Steroid hormone-like drugs including CPA and flutamide are known to potentially induce hepatotoxicity when administered at high doses (*Rojas et al., 2020*). Following administration, reactive metabolites of these drugs are formed, which may lead to hepatitis (*Giorgetti et al., 2017*; *Kassid, Odhaib & Altemimi, 2022*). In humans, various case studies report about hepatotoxicity following treatment with flutamide or CPA (*e.g.*, reviewed by (*Giorgetti et al., 2017*) and (*Kumar et al., 2021*)). The hepatotoxic effect of antiandrogenic substances is also proven by *in vivo* and *in vitro* experiments (*de Gregorio et al., 2021*; *de Gregorio et al., 2016*; *Ding et al., 2021*; *Legendre et al., 2014*; *Leone et al., 2014*; *Snouber et al., 2013*). Therefore, it is traceable that the tested substances adversely affect embryonic development and result in increased mortality rates when administered in higher doses.

Furthermore, *in ovo* exposure to 2 and 20 $\mu$g CPA/g egg resulted in significantly smaller embryos as displayed by shortened lengths of skull, ulna and tarsometatarsus. A number of experiments with mammals suggest that antiandrogenic substances can affect bone

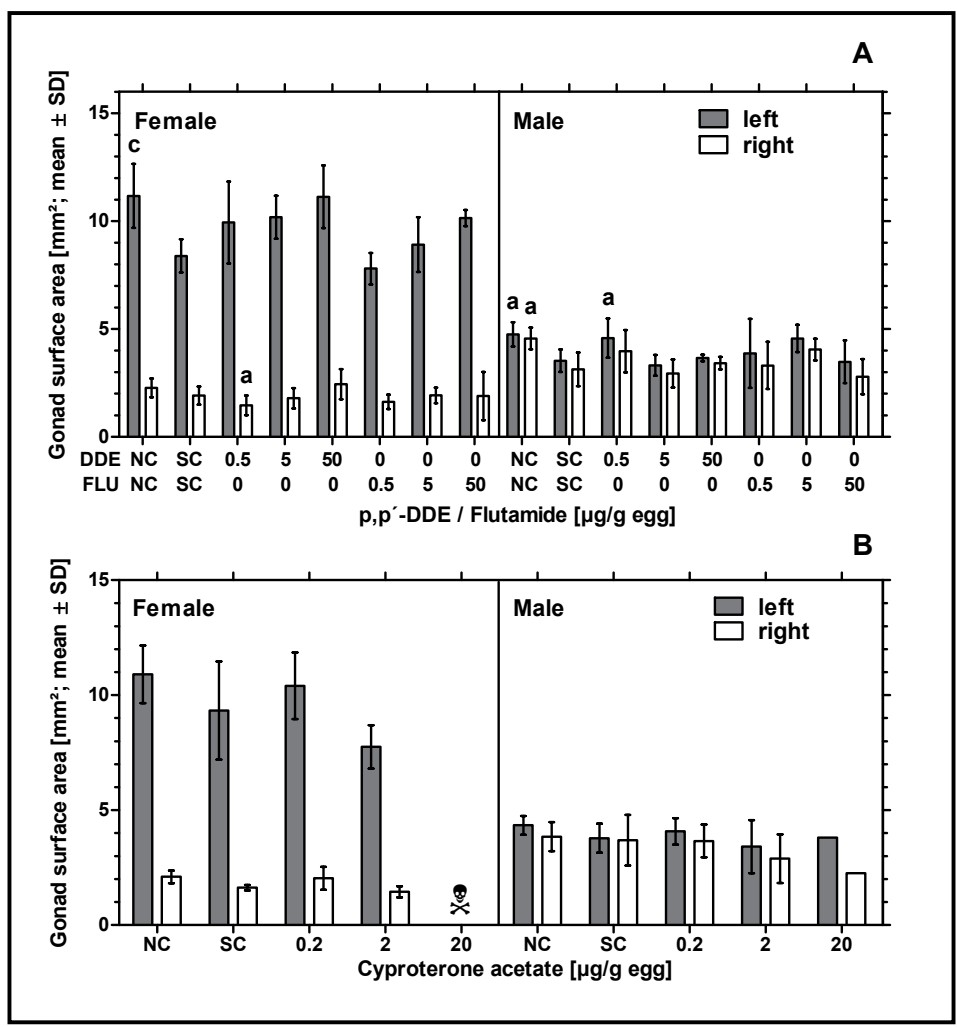

**Figure 3** Effects of *in ovo* exposure to *p,p'*- DDE and flutamide (A) or cyproterone acetate (B) on left and right gonad surface area of embryos of the domestic fowl (*Gallus gallus domesticus*) on embryonic day 19. Administered doses: *p,p'*- DDE (DDE): 0.5, 5, 50 µg/g egg; flutamide (FLU): 0.5, 5, 50 µg/g egg; cyproterone acetate (CPA): 0.2, 2, 20 µg/g egg. Statistical analysis by one-way ANOVA with Dunnett's multiple comparisons test. NC: untreated control group. Lowercase indicates significant differences compared to the solvent control (SC). Level of significance: a, $p < 0.05$; c, $p < 0.001$. Skull symbol: high mortality in the test group resulted in an absence of usable gonad tissue for measurements.

maturation and elongation resulting in an extension of the growth phase. *Neumann (1982)* and *Neumann & Topert (1986)* state that antiandrogens act in all target organs for androgens and principally affect all functions which are influenced by androgens. Some of these effects, such as the delay of puberty, inhibition of spermatogenesis, the loss of libido or the atrophy of accessory glands, are more sex-specific, while other effects such as delayed bone maturation or the inhibition of body weight development are less sex-specific. Especially under the influence of CPA, a retardation of bone maturation and longitudinal growth is shown in experiments with rodents (*Hertel, Kramer & Neumann,*

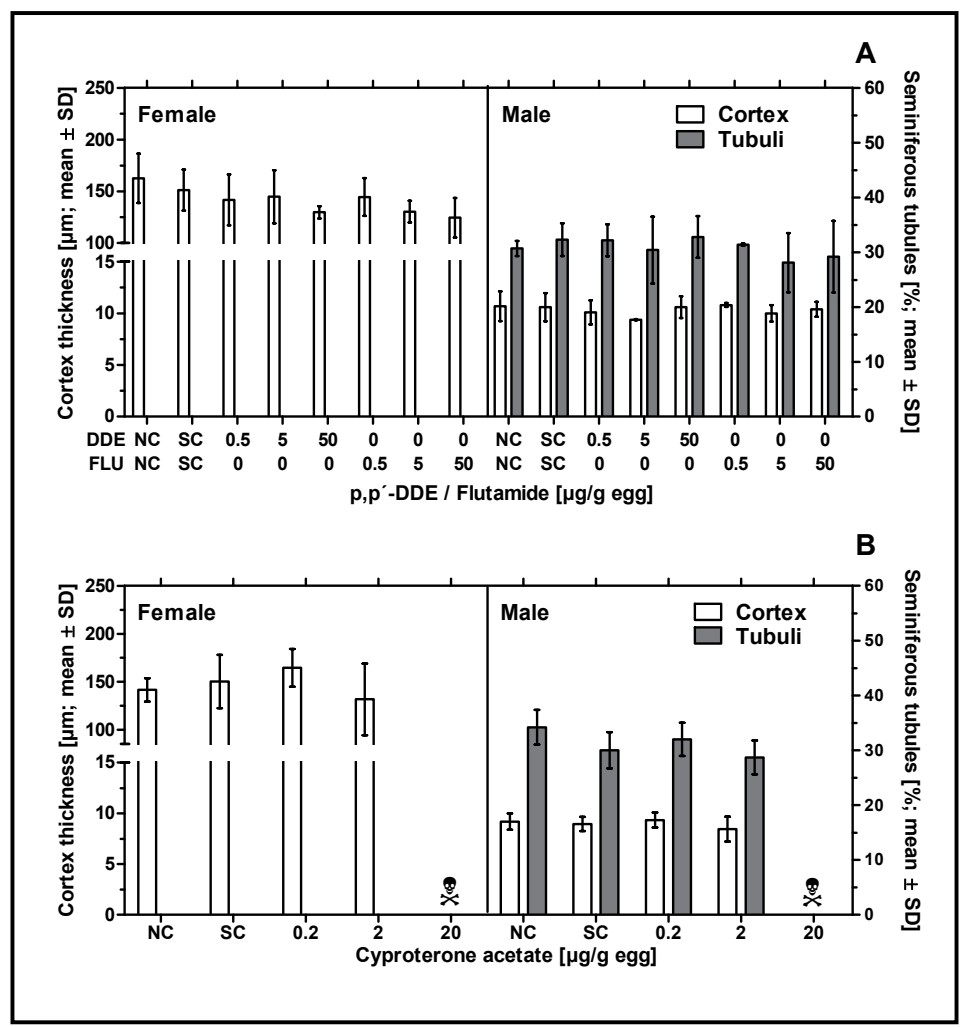

**Figure 4** Effects of *in ovo* exposure to *p,p'*- DDE and flutamide (A) or cyproterone acetate (B) on gonad-based endpoints of embryos of the domestic fowl (*Gallus gallus domesticus*) on embryonic day 19. Administered doses: *p,p'*- DDE (DDE): 0.5, 5, 50 μg/g egg; flutamide (FLU): 0.5, 5, 50 μg/g egg; cyproterone acetate (CPA): 0.2, 2, 20 μg/g egg. Endpoints shown: gonadal cortex thickness (males and females) and percentage of seminiferous tubules (males) of the left gonad. Statistical analysis by one-way ANOVA with Dunnett's multiple comparisons test. NC: untreated control group. SC: solvent control group. Skull symbol: high mortality in the test group resulted in an absence of usable gonad tissue for measurements.

*1969*; *Schenck & Neumann, 1973*). This coincides with the findings of the present study. Delayed embryonic growth of *G. gallus domesticus* can thus be directly attributed to the treatment with CPA. This suggests that the chick embryo is a suitable test system for the identification of substance-related mortality and developmental delays.

Embryonic energy reserves were determined to investigate the impact of CPA treatment on embryonic development. With increasing concentrations of CPA, a significant decrease in the levels of glycogen and protein was observed. However, 20 μg CPA/g egg resulted

in a significantly increased content of lipids. As CPA is known to potentially induce hepatotoxicity (*Kumar et al., 2021*; *Leone et al., 2014*) it can be suspected that higher contents of lipids in the liver of embryos of *G. gallus domesticus* are signs of an incipient liver damage. In reverse, lower contents of glycogen could be a result of increased metabolic activity for detoxification.

In summary, the present results show that the morphometric endpoints studied are well suited for the detection of the toxic effects of selected antiandrogenic model compounds on embryonic development of *G. gallus domesticus*.

## Morphological observation of the gonads—gonad surface area and left testis and ovary

*In ovo* exposure to flutamide or CPA did not affect male and female gonad surface areas. *In ovo* exposure to 0.5 µg *p,p'*- DDE/g egg resulted in significantly smaller left male and right female gonad surface areas. However, we assume that these results are due to the small number of embryos analyzed per experimental group.

The significant difference in gonad surface area between the untreated and the solvent control as found in the experiment with flutamide and *p,p'*- DDE confirms the previous findings of our project group. In *Jessl, Scheider & Oehlmann (2018)* we intensively analyzed untreated and solvent control groups and found that treatment with DMSO resulted in reduced gonad surface areas in both sexes. Gonad surface areas of these gonads decreased with increasing volume of the solvent. Although we cannot clarify the cause of this effect, we suspect a growth-inhibiting effect caused by the low basic toxicity of the solvent or a possible endocrine-mediated effect of the solvent (*Jessl, Scheider & Oehlmann, 2018*).

Summarized, the present study shows that the antiandrogens flutamide, *p,p'*- DDE and CPA have no effects on the tested endpoint gonad surface area. This raises the question of whether the gonads of *G. gallus domesticus* are target organs for antiandrogenic substances. It is known that in mammals, antiandrogens principally affect all androgen-dependent functions and organ systems. In rats, AR antagonists such as flutamide, *p,p'*- DDE and CPA are known to be potent inhibitors of androgen dependent reproductive organs (*Neri & Peets, 1975*) resulting in reduced anogenital distance (*Fussell et al., 2015*; *Pallares et al., 2014*), hypospadias (*Sinclair et al., 2017*), atrophy of seminal vesicles (*Pallares et al., 2014*), nipple retention (*Fussell et al., 2015*), delayed onset of puberty and reduced ventral prostate weight in male rats (*Kelce et al., 1995*). In fish, flutamide adversely affects male and female sex differentiation. In females it causes hastened ovarian development with distorted morphology (*Chakrabarty et al., 2012*), a reduction of relative gonads size (*Milsk et al., 2016*) and disturbs female reproduction (*Bhatia et al., 2014b*). In males, flutamide affects secondary sex characteristics (*Milsk et al., 2016*) and testicular growth (*Bhatia & Kumar, 2016*; *Bhatia et al., 2014a*; *Yin et al., 2017*).

Also, in birds antiandrogens are known for their hormonal disruptive potential with DDT and its derivates as the most popular representatives. Enriched through the food chain, this compound leads to egg shell thinning in seabirds (*Fry & Toone, 1981*; *Hickey & Anderson, 1968*) and intersex testes and oviducts in gull embryos (*Fry & Toone, 1981*). While many species of raptorial and fish-eating birds are shown to be highly sensitive

to DDT-related eggshell thinning, other species such as chicken and quail are almost completely insensitive to this end point (*Peakall & Lincer, 1996*).

Considering gonadal endpoints, antiandrogenic effects may be quite different, depending on the test substance and the species used. *O,p'*- DDT for example adversely affects the gonads of domestic roosters resulting in cloacal defects, deformations of one or both testes and smaller diameters of seminiferous tubules (*Blomqvist et al., 2006*). In quail *o,p'*- DDT leads to a significant reduction of plasma testosterone levels and the area of the cloacal gland while testis weight and diameter of seminiferous tubules are not affected. Ovaries appear unaffected although the right oviduct is regressed and the left oviduct is shortened (*Halldin et al., 2003*). *Quinn, Summitt & Ottinger (2008)* report that *p,p'*- DDE has no significant effect on gonadal physiology and morphology in both sexes of quail. *Kamata et al. (2013)* studied the effects of *p,p'*- DDE and *p,p'*- DDT on avian reproduction and did not find any changes in the morphology of reproductive organs, which coincides with the results of the present study. As *p,p'*- DDE is suspected to have antiandrogenic and estrogenic modes of action (*Burgos-Aceves et al., 2021*; *Hoffmann & Kloas, 2016*; *Muñoz-de Toro et al., 2006*), the results of the present study were also considered with regard to potential estrogenic effects on gonadal sex differentiation of the chicken embryo of *G. gallus domesticus*. Exposure of male avian embryos to estrogen and estrogen-active EDCs, classically results in the effective feminization of the reproductive organs (*Berg et al., 2019*; *Berg, Halldin & Brunstrom, 2001*; *Intarapat, Sailasuta & Satayalai, 2014*; *Mentor et al., 2020b*; *Scheib & Reyssbrion, 1979*; *Shibuya et al., 2004*). The most frequent change in male reproductive organ differentiation is the formation of ovotestes, as observed in domestic chicken embryos and other avian species (*Ellis et al., 2012*; *Esener & Bozkurt, 2018*; *Mattsson, Olsson & Brunstrom, 2008*; *Shibuya et al., 2005*). These results are consistent with our own studies investigating *in ovo* exposure of chicken embryos to different estrogenic compounds. In *Jessl, Scheider & Oehlmann (2018)*; *Jessl et al. (2018)*) we found that embryonic males of *G. gallus domesticus* responded with the formation of left ovotestes when exposed to 17 $\alpha$-ethinylestradiol (EE$_2$) or bisphenol A (BPA), even at very low doses. At the morphological level, the administration of EE$_2$ led to the formation of ovotestes that resembled the ovaries of untreated female embryos in size and appearance. At the histological level, both, EE$_2$ and BPA, resulted in a concentration-dependent, significant increase in left testicular cortex thickness. The medulla of the left testis was partially interspersed with *lacunae*, comparable with the histological appearance of left ovaries of untreated females. Furthermore, the percentage of seminiferous tubules of the left testis significantly decreased with the administration of EE$_2$. The results of the present study do not support the theory of a dominant estrogenic effect of *p,p'*- DDE. None of the effects mentioned above was found in the previous study. The absence of such estrogen-related effects indicate that *p,p'*-DDE has no estrogenic activity in the chicken embryo at the doses tested. Our findings are further supported by the study of *Kamata et al. (2013)*, who stated that *p,p'*- DDE theoretically has the potential to activate the ER of domestic fowl, but doses used in their study did not affect the reproductive organ morphology of Japanese quail.

*Wollman & Hamilton (1968)* and *Wollman & Hamilton (1967)* for example demonstrate an inhibitory effect of CPA on comb size of chicks explained by the antagonization of

androgenic effects. Furthermore, *Utsumi & Yoshimura (2009)* described CPA to have inhibitory effects on the development of cloacal gland structures in quail. Since AR and mRNA are produced in this tissue, cloacal glands are target organs for androgens. On the contrary, CPA did not cause significant structural differences in quail ovaries and testes. In quail, flutamide-treatment affects male copulatory behavior as demonstrated by reduced TP-activated strutting representing a central nervous system effect of the drug (*Adkinsregan & Garcia, 1986*). However, in the chicken embryo, flutamide-treatment leads to defects of sex cord formation and *lacunae* in the developing left ovary (*Katoh, Ogino & Yamada, 2006*).

The lack of antiandrogenic effects on gonad-based endpoint seems to be related to the avian hormonal system. In birds, sexual differentiation is dependent on estrogen (*Brunstrom et al., 2009*; *Vaillant et al., 2001b*). The presence of estrogen causes the differentiation toward the female sex, whereas the absence of estrogen causes the differentiation toward the male sex. It is known, that AR is expressed in male and female chicken gonads throughout embryonic development (*Katoh, Ogino & Yamada, 2006*; *Tanaka, Izumi & Kuroiwa, 2017*). Thus, binding of the investigated compounds CPA, flutamide, or *p,p'*- DDE to the AR and disruption of subsequent AR signaling is potentially possible. Throughout embryogenesis AR expression is markedly higher in female gonads than in male gonads. In females, AR is localized in the left ovary, especially in cells of lacunae within the medulla, and in cortical cords within the cortex. In contrast, such nuclear localized signals could not be detected in male testes, although AR proteins were distributed in the developing testicular cords (*Katoh, Ogino & Yamada, 2006*; *Tanaka, Izumi & Kuroiwa, 2017*). This shows that androgen is mainly relevant for the female sex, as it provides the substrate for aromatization and estrogen production (*Ayers, Sinclair & Smith, 2013*; *Estermann, Major & Smith, 2021*; *Scheider et al., 2014*) and thus is indispensable for the correct formation of the female gonads. In the male sex there is no evidence that androgen plays a relevant role in testicular morphogenesis (*Estermann, Major & Smith, 2021*; *Groenendijk-Huijbers & Van Schaik, 1976*).

If the proposed chicken embryo-based test system is to detect the effects of androgen antagonists in the future, further endpoints should be considered, like the inhibitory effects of different (anti)androgens on *bursa fabricii* development which have been studied in quail and chicken. The *bursa fabricii*, an avian lymphoid organ located in the cloacal region, theoretically represents a primary target for antiandrogen active EDCs due to its sensitivity to androgens (*Quinn & Ottinger, 2006*). The *bursa fabricii* experiences significant weight loss upon administration of androgen such as testosterone propionate, whereas simultaneous administration of an antiandrogen such as flutamide can reverse androgen-induced bursal regression (*Burke, 1996*; *Henry & Burke, 1999*; *Utsumi & Yoshimura, 2009*). In the study by *Katoh, Ogino & Yamada (2006)*, *in ovo* administration of flutamide to embryonic chicken results in defects of sex cord formation and *lacunae* in the developing left ovary. These endpoints, which are very similar to the endpoint cortex thickness investigated in the present work, are easy to evaluate and could be readily integrated into the existing analysis workflow.

Taken together, *in ovo* exposure of *G. gallus domesticus* to the selected antiandrogenic model compounds CPA, flutamide, and *p,p'*- DDE results in no teratogenic effects on male and female gonads of E19 embryos. The selected gonad-based endpoints in embryos of *G. gallus domesticus* seem to be insensitive to antiandrogens as they are not necessarily target organs/tissues for androgens. However, this does not mean that the gonad-based endpoints studied are generally useless for the investigation of the effects of potential EDCs on embryonic sexual differentiation. In further investigations we found that especially estrogens but also antiestrogens and androgens can adversely affect embryonic sexual differentiation of *G. gallus domesticus*. In *Jessl, Scheider & Oehlmann (2018)* and *Jessl et al. (2018)*, we have shown that *in ovo* exposure of chick embryos to EE$_2$, a synthetic estrogen, resulted in a distinct feminization of genetic males which formed female-like cortex tissue in their left gonads. In addition, EE$_2$ treatment resulted in a reduction of the percentage of seminiferous tubules. In *Jessl et al. (2018)* we demonstrated that the antiestrogen tamoxifen affected female embryonic sex differentiation and caused a size reduction of the left ovary and malformations of the ovarian cortex. In *Scheider et al. (2018)* we investigated the effects of the functional androgen tributyltin (TBT) and found it to affect sex differentiation as it led to virilization effects of female embryos which were mainly characterized by a significant reduction of the left cortex. In conclusion, it can be stated, that the proposed test system is able to detect adverse effects of (anti)estrogens and androgens on gonad-based endpoints, as shown in our previous studies (*Jessl, Scheider & Oehlmann, 2018*; *Jessl et al., 2018*; *Scheider et al., 2018*) while it finds its limitation with antiandrogenic compounds, as shown with CPA, flutamide, and *p,p'*- DDE investigated in the present work.

## OVERALL CONCLUSIONS

The focus of the present work was to study the effects of the antiandrogenic compounds CPA, *p,p'*- DDE and flutamide on embryonic gonadal sex differentiation of chicken (*Gallus gallus domesticus*). *In ovo* exposure to all three substances had no effects on gonadal endpoints. In contrast to test results with estrogenic, antiestrogenic and androgenic compounds, these endpoints were not affected by antiandrogenic EDCs and are therefore shown no suitable parameters for the detection of chemicals with antiandrogenic properties in chick embryos.

However, *in ovo* exposure to CPA resulted in significantly smaller embryos than in control groups as displayed by shortened lengths of skull, ulna and tarsometatarsus. This suggests that the chick embryo is a suitable test system for the identification of substance-related mortality and developmental delays.

## ACKNOWLEDGEMENTS

We thank Andrea Dombrowski, Simone Ziebart, Alina Helmes and Katrin Collmar for technical assistance.

## Funding

This work was carried out in the framework of the project GenOvotox II, funded by the Federal Ministry of Education and Research (BMBF; project no 031A104B). There was no additional external funding received for this study. The funders had no role in study design, data collection and analysis, decision to publish, or preparation of the manuscript.

## Grant Disclosures

The following grant information was disclosed by the authors:
GenOvotox II:  031A104B.

## Competing Interests

Prof. Dr Jörg Oehlmann is an Academic Editor for PeerJ. Luzie Jessl is an employee of R-Biopharm AG.

## Author Contributions

- Luzie Jessl conceived and designed the experiments, performed the experiments, analyzed the data, prepared figures and/or tables, authored or reviewed drafts of the article, and approved the final draft.
- Jörg Oehlmann conceived and designed the experiments, analyzed the data, authored or reviewed drafts of the article, and approved the final draft.

## Data Availability

 The raw measurements are available in the Supplementary File.

## Supplemental Information

Supplemental information for this article can be found online at http://dx.doi.org/10.7717/peerj.16249#supplemental-information.

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

# PeerJ

in human and wildlife: a possible implication of mitochondria. *Environmental Toxicology and Pharmacology* **87**:103684 DOI 10.1016/j.etap.2021.103684.

**Burke WH. 1996.** Effects of an *in ovo* injection of an anti-androgen on embryonic and posthatching growth of broiler chicks. *Poultry Science* **75(5)**:648–655 DOI 10.3382/ps.0750648.

**Chakrabarty S, Rajakumar A, Raghuveer K, Sridevi P, Mohanachary A, Prathibha Y, Bashyam L, Dutta-Gupta A, Senthilkumaran B. 2012.** Endosulfan and flutamide, alone and in combination, target ovarian growth in juvenile catfish, Clarias batrachus. *Comparative Biochemistry and Physiology C-Toxicology & Pharmacology* **155(3)**:491–497 DOI 10.1016/j.cbpc.2011.12.007.

**Chen CC, Liu YS, Cheng CC, Wang CL, Liao MH, Tseng CN, Chang HW. 2012.** High-throughput sex identification by melting curve analysis in blue-breasted quail and chicken. *Theriogenology* **77(9)**:1951–1958 DOI 10.1016/j.theriogenology.2011.12.004.

**Davies IM, Harding MJC, Bailey SK, Shanks AM, Lange R. 1997.** Sublethal effects of tributyltin oxide on the dogwhelk *Nucella lapillus*. *Marine Ecology Progress Series* **158**:191–204 DOI 10.3354/meps158191.

**de Gregorio LS, Franco-Belussi L, Goldberg J, De Oliveira C. 2021.** Nonylphenol and cyproterone acetate effects in the liver and gonads of *Lithobates catesbeianus* (Anura) tadpoles and juveniles. *Environmental Science and Pollution Research* **28(44)**:62593–62604 DOI 10.1007/s11356-021-14599-7.

**de Gregorio LS, Franco-Belussi L, Gomes FR, de Oliveira C. 2016.** Flutamide effects on morphology of reproductive organs and liver of neotropical Anura, Rhinella schneideri. *Aquatic Toxicology* **176**:181–189 DOI 10.1016/j.aquatox.2016.04.022.

**Delbes G, Blazquez M, Fernandino JI, Grigorova P, Hales BF, Metcalfe C, Navarro-Martin L, Parent L, Robaire B, Rwigemera A, Vander Kraak G, Wade M, Marlatt V. 2022.** Effects of endocrine disrupting chemicals on gonad development: mechanistic insights from fish and mammals. *Environmental Research* **204**:112040 DOI 10.1016/j.envres.2021.112040.

**Ding YN, Ma HH, Xu YS, Yang F, Li Y, Shi FG, Lu YF. 2021.** Potentiation of flutamide-induced hepatotoxicity in mice by Xian-Ling-Gu-Bao through induction of CYP1A2. *Journal of Ethnopharmacology* **278**:114299 DOI 10.1016/j.jep.2021.114299.

**Eising CM, Eikenaar C, Schwbl H, Groothuis TGG. 2001.** Maternal androgens in black-headed gull (*Larus ridibundus*) eggs: consequences for chick development. *Proceedings of the Royal Society B-Biological Sciences* **268(1469)**:839–846 DOI 10.1098/rspb.2001.1594.

**Ellis HL, Shioda K, Rosenthal NF, Coser KR, Shioda T. 2012.** Masculine epigenetic sex marks of the CYP19A1/aromatase promoter in genetically male chicken embryonic gonads are resistant to estrogen-induced phenotypic sex conversion. *Biology of Reproduction* **87(1)**:1–12 DOI 10.1093/biolreprod/87.s1.1.

**Esener OBB, Bozkurt HH. 2018.** Effects of *in ovo* injected bisphenol A on the testis of one day old chickens. *Ankara Universitesi Veteriner Fakultesi Dergisi* **65(1)**:21–28 DOI 10.1501/Vetfak_0000002823.

**Estermann MA, Major AT, Smith CA. 2021.** Genetic regulation of avian testis development. *Genes* **12(9)**:1459 DOI 10.3390/genes12091459.

**Farhat A, Crump D, Bidinosti L, Boulanger E, Basu N, Hecker M, Head JA. 2020.** An early-life stage alternative testing strategy for assessing the impacts of environmental chemicals in birds. *Environmental Toxicology and Chemistry* **39(1)**:141–154 DOI 10.1002/etc.4582.

**Fitzgerald JA, Trznadel M, Katsiadaki I, Santos EM. 2020.** Hypoxia modifies the response to flutamide and linuron in male three-spined stickleback (*Gasterosteus aculeatus*). *Environmental Pollution* **263**:114326 DOI 10.1016/j.envpol.2020.114326.

**Fridolfsson AK, Ellegren H. 1999.** A simple and universal method for molecular sexing of non-ratite birds. *Journal of Avian Biology* **30(1)**:116–121 DOI 10.2307/3677252.

**Fry DM, Toone CK. 1981.** DDT-induced feminization of gull embryos. *Science* **213(4510)**:922–924 DOI 10.1126/science.7256288.

**Fussell KC, Schneider S, Buesen R, Groeters S, Strauss V, Melching-Kollmuss S, van Ravenzwaay B. 2015.** Investigations of putative reproductive toxicity of low-dose exposures to flutamide in Wistar rats. *Archives of Toxicology* **89(12)**:2385–2402 DOI 10.1007/s00204-015-1622-6.

**Gasc JM, Stumpf WE. 1981.** Sexual-differentiation of the urogenital tract in the chicken-embryo - autoradiographic localization of sex-steroid target-cells during development. *Journal of Embryology and Experimental Morphology* **63(1)**:207–223.

**Giorgetti R, Di Muzio M, Giorgetti A, Girolami D, Borgia L, Tagliabracci A. 2017.** Flutamide-induced hepatotoxicity: ethical and scientific issues. *European Review for Medical and Pharmacological Sciences* **21**:69–77.

**Gismondi E, Cauchie HM, Cruciani V, Joaquim-Justo C. 2019.** Targeted impact of cyproterone acetate on the sexual reproduction of female rotifers. *Ecotoxicology* **28(6)**:643–649 DOI 10.1007/s10646-019-02063-9.

**Gooding MP, Wilson VS, Folmar LC, Marcovich DT, LeBlanc GA. 2003.** The biocide tributyltin reduces the accumulation of testosterone as fatty acid esters in the mud snail (*Ilyanassa obsoleta*). *Environmental Health Perspectives* **111(4)**:426–430 DOI 10.1289/ehp.5779.

**Groenendijk-Huijbers M, Van Schaik J. 1976.** Effects of hemicastration, testis implantation and administration of testosterone propionate on the female embryonic genital tract in various breeds and strains of chickens. *Verhandlungen der Anatomischen Gesellschaft* **1976**:179–182.

**Guioli S, Zhao DB, Nandi S, Clinton M, Lovell-Badge R. 2020.** Oestrogen in the chick embryo can induce chromosomally male ZZ left gonad epithelial cells to form an ovarian cortex that can support oogenesis. *Development* **147(4)**:181693.

**Halldin K, Holm L, Ridderstrale Y, Brunstrom B. 2003.** Reproductive impairment in Japanese quail (*Coturnix japonica*) after *in ovo* exposure to *o, p'*- DDT. *Archives of Toxicology* **77(2)**:116–122 DOI 10.1007/s00204-002-0417-8.

**Hamburger V, Hamilton HL. 1992.** A series of normal stages in the development of the chick-embryo (reprinted from Journal of Morphology, 88, 1951). *Developmental Dynamics* **195(4)**:231–272 DOI 10.1002/aja.1001950404.

**Harnett KG, Moore LG, Chin A, Cohen IC, Lautrup RR, Schuh SM. 2021.** Terato-genicity and toxicity of the new BPA alternative TMBPF, and BPA, BPS, and BPAF in chick embryonic development. *Current Research in Toxicology* **2**:399–410 DOI 10.1016/j.crtox.2021.11.001.

**Heinlein CA, Chang C. 2004.** Androgen receptor in prostate cancer. *Endocrine Reviews* **25(2)**:276–308 DOI 10.1210/er.2002-0032.

**Henry MH, Burke WH. 1999.** The effects of *in ovo* administration of testosterone or an antiandrogen on growth of chick embryos and embryonic muscle characteristics. *Poultry Science* **78(7)**:1006–1013 DOI 10.1093/ps/78.7.1006.

**Hertel P, Kramer M, Neumann F. 1969.** Effect of an antiandrogen (cyproterone acetate) on bone growth and bone maturation in male rats. *Arzneimittel-Forschung* **19(11)**:1777–1790.

**Hickey JJ, Anderson DW. 1968.** Chlorinated hydrocarbons and eggshell changes in raptorial and fish-eating birds. *Science* **162(3850)**:271–273 DOI 10.1126/science.162.3850.271.

**Ho V, Pelland-St-Pierre L, Gravel S, Bouchard MF, Verner MA, Labreche F. 2022.** Endocrine disruptors: challenges and future directions in epidemiologic research. *Environmental Research* **204**:111969 DOI 10.1016/j.envres.2021.111969.

**Hoffmann F, Kloas W. 2016.** p,p ′-Dichlordiphenyldichloroethylene (p, p ′-DDE) can elicit antiandrogenic and estrogenic modes of action in the amphibian *Xenopus laevis*. *Physiology & Behavior* **167**:172–178 DOI 10.1016/j.physbeh.2016.09.012.

**Holm L, Blomqvist A, Brandt I, Brunstrom B, Ridderstrale Y, Berg C. 2006.** Embryonic exposure to *o, p*'- DDT causes eggshell thinning and altered shell gland carbonic anhydrase expression in the domestic hen. *Environmental Toxicology and Chemistry* **25(10)**:2787–2793 DOI 10.1897/05-619R.1.

**Intarapat S, Sailasuta A, Satayalai O. 2014.** Anatomical and histological changes of reproductive organs in Japanese quail (*Coturnix japonica*) embryos after *in ovo* exposure to genistein. *International Journal of Poultry Science* **13(1)**:1–13.

**Ioannidis J, Taylor G, Zhao DB, Liu L, Idoko-Akoh A, Gong DQ, Lovell-Badge R, Guioli S, McGrew MJ, Clinton M. 2021.** Primary sex determination in birds depends on DMRT1 dosage, but gonadal sex does not determine adult secondary sex characteristics. *Proceedings of the National Academy of Sciences of the United States of America* **118(10)**:e2020909118.

**Jessl L, Lenz R, Massing FG, Scheider J, Oehlmann J. 2018.** Effects of estrogens and antiestrogens on gonadal sex differentiation and embryonic development in the domestic fowl (*Gallus gallus domesticus*). *PeerJ* **6**:e5094 DOI 10.7717/peerj.5094.

**Jessl L, Scheider J, Oehlmann J. 2018.** The domestic fowl (*Gallus gallus domesticus*) embryo as an alternative for mammalian experiments –Validation of a test method for the detection of endocrine disrupting chemicals. *Chemosphere* **196**:502–513 DOI 10.1016/j.chemosphere.2017.12.131.

**Jin SC, Shao L, Song XP, Xiao JH, Ouyang K, Zhang KL, Yang JX. 2019.** Fertilization and male fertility in the rotifer *Brachionus calyciflorus* in the presence of three

environmental endocrines. *Chemosphere* **220**:146–154
DOI 10.1016/j.chemosphere.2018.12.097.

**Kamata R, Shiraishi F, Nakamura K. 2020.** Avian eggshell thinning caused by transo-
varian exposure to *o, p*'- DDT: Changes in histology and calcium-binding protein
production in the oviduct uterus. *Journal of Toxicological Sciences* **45(3)**:131–136
DOI 10.2131/jts.45.131.

**Kamata R, Shiraishi F, Takahashi S, Shimizu A, Nakajima D, Kageyama S, Sasaki
T, Temma K. 2013.** The effect of transovarian exposure to *p, p*'- DDT and *p, p*'-
DDE on avian reproduction using Japanese quails. *Journal of Toxicological Sciences*
**38(6)**:903–912 DOI 10.2131/jts.38.903.

**Kassid OM, Odhaib SA, Altemimi MT. 2022.** Flutamide-induced hepatotoxicity: a case
report. *Journal of Biological Research - Bollettino della Società Italiana di Biologia
Sperimentale* **95(2)**:10371.

**Katoh H, Ogino Y, Yamada G. 2006.** Cloning and expression analysis of andro-
gen receptor gene in chicken embryogenesis. *Febs Letters* **580(6)**:1607–1615
DOI 10.1016/j.febslet.2006.01.093.

**Keibel F, Abraham K. 1900.** *Normentafel zur Entwicklungsgeschichte des Huhnes, Gallus
domesticus.* Jena: Fischer.

**Kelce WR, Stone CR, Laws SC, Gray LE, Kemppainen JA, Wilson EM. 1995.** Persistent
DDT metabolite *p, p*'-DDE is a potent androgen receptor antagonist. *Nature*
**375(6532)**:581–585 DOI 10.1038/375581a0.

**Kumar P, Reddy S, Kulkarni A, Sharma M, Rao PN. 2021.** Cyproterone acetate-induced
acute liver failure: a case report and review of the literature. *Journal of Clinical and
Experimental Hepatology* **11(6)**:739–741 DOI 10.1016/j.jceh.2021.01.003.

**Legendre A, Jacques S, Dumont F, Cotton J, Paullier P, Fleury MJ, Leclerc E. 2014.**
Investigation of the hepatotoxicity of flutamide: pro-survival/apoptotic and necrotic
switch in primary rat hepatocytes characterized by metabolic and transcriptomic
profiles in microfluidic liver biochips. *Toxicology in Vitro* **28(5)**:1075–1087
DOI 10.1016/j.tiv.2014.04.008.

**Leone A, Nie A, Parker JB, Sawant S, Piechta LA, Kelley MF, Kao LM, Proctor
SJ, Verheyen G, Johnson MD, Lord PG, McMillian MK. 2014.** Oxidative
stress/reactive metabolite gene expression signature in rat liver detects idiosyn-
cratic hepatotoxicants. *Toxicology and Applied Pharmacology* **275(3)**:189–197
DOI 10.1016/j.taap.2014.01.017.

**Marlatt VL, Bayen S, Castaneda-Cortes D, Delbes G, Grigorova P, Langlois VS,
Martyniuk CJ, Metcalfe CD, Parent L, Rwigemera A, Thomson P, Vander Kraak
G. 2022.** Impacts of endocrine disrupting chemicals on reproduction in wildlife and
humans. *Environmental Research* **208**:112584 DOI 10.1016/j.envres.2021.112584.

**Mattsson A, Olsson JA, Brunstrom B. 2008.** Selective estrogen receptor *α* activation
disrupts sex organ differentiation and induces expression of vitellogenin II and
very low-density apolipoprotein II in Japanese quail embryos. *Reproduction*
**136(2)**:175–186 DOI 10.1530/REP-08-0100.

**McAllister BG, Kime DE. 2003.** Early life exposure to environmental levels of the aromatase inhibitor tributyltin causes masculinisation and irreversible sperm damage in zebrafish (*Danio rerio*). *Aquatic Toxicology* **65(3)**:309–316 DOI 10.1016/S0166-445X(03)00154-1.

**Mentessidou A, Salakos C, Chrousos G, Kanaka-Gantenbein C, Kostakis A, Mirilas P. 2021.** Morphologic alterations of the genital mesentery implicated in testis non-descent in rats prenatally exposed to flutamide. *Andrology* **9(1)**:440–450 DOI 10.1111/andr.12903.

**Mentor A, Bornehag C-G, Jönsson M, Mattsson A. 2020a.** A suggested bisphenol A metabolite (MBP) interfered with reproductive organ development in the chicken embryo while a human-relevant mixture of phthalate monoesters had no such effects. *Journal of Toxicology and Environmental Health, Part A* **83(2)**:66–81 DOI 10.1080/15287394.2020.1728598.

**Mentor A, Wänn M, Brunström B, Jönsson M, Mattsson A. 2020b.** Bisphenol AF and bisphenol F induce similar feminizing effects in chicken embryo testis as bisphenol A. *Toxicological Sciences* **178(2)**:239–250 DOI 10.1093/toxsci/kfaa152.

**Metcalfe CD, Bayen S, Desrosiers M, Munoz G, Sauve S, Yargeau V. 2022.** Methods for the analysis of endocrine disrupting chemicals in selected environmental matrixes. *Environmental Research* **206**:112616 DOI 10.1016/j.envres.2021.112616.

**Milsk R, Cavallin JE, Durhan EJ, Jensen KM, Kahl MD, Makynen EA, Martinovic-Weigelt D, Mueller N, Schroeder A, Villeneuve DL, Ankley GT. 2016.** A study of temporal effects of the model anti-androgen flutamide on components of the hypothalamic-pituitary-gonadal axis in adult fathead minnows. *Aquatic Toxicology* **180**:164–172 DOI 10.1016/j.aquatox.2016.09.021.

**Neri RO, Peets EA. 1975.** Biological aspects of antiandrogens. *Journal of Steroid Biochemistry* **6(6)**:815–819 DOI 10.1016/0022-4731(75)90309-X.

**Neumann F. 1982.** Pharmacology and clinical use of antiandrogens - a short review. *Irish Journal of Medical Science* **151(3)**:61–70 DOI 10.1007/BF02940148.

**Neumann F, Topert M. 1986.** Pharmacology of antiandrogens. *Journal of Steroid Biochemistry and Molecular Biology* **25(5B)**:885–895 DOI 10.1016/0022-4731(86)90320-1.

**OECD. 2007.** *OECD guideline for the testing of chemicals (440) Uterotrophic bioassay in rodents.* Paris: Organisation for Economic Co-operation and Development.

**OECD. 2009.** *OECD guideline for the testing of chemicals (441) Hershberger bioassay in rats.* Paris: Organisation for Economic Co-operation and Development.

**Ottinger MA, Lavoie E, Thompson N, Barton A, Whitehouse K, Barton M, Abdelnabi M, Quinn M, Panzica G, Viglietti-Panzica C. 2008.** Neuroendocrine and behavioral effects of embryonic exposure to endocrine disrupting chemicals in birds. *Brain Research Reviews* **57(2)**:376–385 DOI 10.1016/j.brainresrev.2007.08.011.

**Pallares ME, Adrover E, Imsen M, Gonzalez D, Fabre B, Mesch V, Baier CJ, Antonelli MC. 2014.** Maternal administration of flutamide during late gestation affects the brain and reproductive organs development in the rat male offspring. *Neuroscience* **278**:122–135 DOI 10.1016/j.neuroscience.2014.07.074.

**Paradisi R, Fabbri R, Battaglia C, Venturoli S. 2013.** Ovulatory effects of flutamide in the polycystic ovary syndrome. *Gynecological Endocrinology* **29(4)**:391–395 DOI 10.3109/09513590.2012.754876.

**Peakall DB, Lincer JL. 1996.** Do PCBs cause eggshell thinning? *Environmental Pollution* **91(1)**:127–129 DOI 10.1016/0269-7491(95)00012-G.

**Quinn JMJ, Ottinger MA. 2006.** Embryonic effects of androgen active endocrine disrupting chemicals on avian immune and reproductive systems. *Journal of Poultry Science* **43(1)**:1–11 DOI 10.2141/jpsa.43.1.

**Quinn MJ, Summitt CL, Ottinger MA. 2008.** Consequences of *in ovo* exposure to *p p'*-DDE on reproductive development and function in Japanese quail. *Hormones and Behavior* **53(1)**:249–253 DOI 10.1016/j.yhbeh.2007.10.004.

**Rangel PL, Sharp PJ, Gutierrez CG. 2006.** Testosterone antagonist (flutamide) blocks ovulation and preovulatory surges of progesterone, luteinizing hormone and oestradiol in laying hens. *Reproduction* **131(6)**:1109–1114 DOI 10.1530/rep.1.01067.

**Ratcliffe DA. 1970.** Changes attributable to pesticides in egg breakage frequency and eggshell thickness in some British birds. *Journal of Applied Ecology* **7(1)**:67–115 DOI 10.2307/2401613.

**Rojas PA, Iglesias TG, Barrera F, Mendez GP, Torres J, San Francisco IF. 2020.** Acute liver failure and liver transplantation secondary to flutamide treatment in a prostate cancer patient. *Urology Case Reports* **33**:101370 DOI 10.1016/j.eucr.2020.101370.

**Rolon S, Huynh C, Guenther M, Gardezi M, Phillips J, Gehrand AL, Raff H. 2019.** The effects of flutamide on the neonatal rat hypothalamic-pituitary-adrenal and gonadal axes in response to hypoxia. *Physiological Reports* **7(24)**:e14318.

**Russell WMS, Burch RL. 1959.** *The principles of humane experimental technique. Reprinted 1992.* Wheathampstead: UFAW.

**Scheib D, Reyssbrion M. 1979.** Feminization of the quail by early diethylstilbestrol treatment - histoenzymological investigations on steroid dehydrogenases in the gonads. *Archives d'Anatomie Microscopique et de Morphologie Experimentale* **68(2)**:85–98.

**Scheider J, Afonso-Grunz F, Hoffmeier K, Horres R, Groher F, Rycak L, Oehlmann J, Winter P. 2014.** Gene expression of chicken gonads is sex- and side-specific. *Sexual Development* **8(4)**:178–191.

**Scheider J, Afonso-Grunz F, Jessl L, Hoffmeier K, Winter P, Oehlmann J. 2018.** Morphological and transcriptomic effects of endocrine modulators on the gonadal differentiation of chicken embryos: the case of tributyltin (TBT). *Toxicology Letters* **284**:143–151 DOI 10.1016/j.toxlet.2017.11.019.

**Schenck B, Neumann F. 1973.** Influence of sexual hormones on bone maturation and bone growth of female rats. *Arzneimittel-Forschung/Drug Research* **23(7)**:887–907.

**Sharin T, Gyasi H, Williams KL, Crump D, O'Brien JM. 2021.** Effects of two bisphenol A replacement compounds, 1, 7-bis (4-hydroxyphenylthio)-3, 5-dioxaheptane and bisphenol AF, on development and mRNA expression in chicken embryos. *Ecotoxicology and Environmental Safety* **215**:112140 DOI 10.1016/j.ecoenv.2021.112140.

**Shibuya K, Mizutani M, Sato K, Itabashi M, Nunoya T. 2005.** Comparative evaluation of sex reversal effects of natural and synthetic estrogens in sex reversal test using

F1 (AWE ×WE) Japanese quail embryos. *Journal of Poultry Science* **42**:119–129 DOI 10.2141/jpsa.42.119.

**Shibuya K, Mizutani M, Wada M, Sato K, Nunoya T. 2004.** A new screening model using F1 (AWExWE) Japanese quail embryo for evaluating sex reversal effects. *Journal of Toxicologic Pathology* **17(4)**:245–252 DOI 10.1293/tox.17.245.

**Sinclair AW, Cao M, Pask A, Baskin L, Cunha GR. 2017.** Flutamide-induced hypospadias in rats: a critical assessment. *Differentiation* **94**:37–57 DOI 10.1016/j.diff.2016.12.001.

**Snouber LC, Bunescu A, Naudot M, Legallais C, Brochot C, Dumas ME, Elena-Herrmann B, Leclerc E. 2013.** Metabolomics-on-a-chip of hepatotoxicity induced by anticancer drug flutamide and its active metabolite hydroxyflutamide using HepG2/C3a microfluidic biochips. *Toxicological Sciences* **132(1)**:8–20 DOI 10.1093/toxsci/kfs230.

**Starck M, Ricklefs R. 1997.** *Avian growth and development: evolution within the altricial-precocial spectrum (vol. 1).* Oxford: Oxford University Press.

**Tanaka R, Izumi H, Kuroiwa A. 2017.** Androgens and androgen receptor signaling contribute to ovarian development in the chicken embryo. *Molecular and Cellular Endocrinology* **443(C)**:114–120 DOI 10.1016/j.mce.2017.01.008.

**Muñoz-de Toro M, Durando M, Beldoménico PM, Beldoménico HR, Kass L, García SR, Luque EH. 2006.** Estrogenic microenvironment generated by organochlorine residues in adipose mammary tissue modulates biomarker expression in ER$\alpha$-positive breast carcinomas. *Breast Cancer Research* **8(4)**:R47 DOI 10.1186/bcr1534.

**Utsumi T, Yoshimura Y. 2009.** Sensitive embryonic endpoints with *in ovo* treatment for detecting androgenic and anti-androgenic effects of chemicals in Japanese quail (*Coturnix japonica*). *Poultry Science* **88(5)**:1052–1059 DOI 10.3382/ps.2008-00326.

**Utsumi T, Yoshimura Y. 2011.** Applicability of lectin histochemistry in a test system with *in ovo* treatment for detecting androgenic and antiandrogenic effects of chemicals in Japanese quail (*Coturnix japonica*). *Poultry Science* **90(1)**:168–174 DOI 10.3382/ps.2010-00629.

**Vaillant S, Dorizzi M, Pieau C, Richard-Mercier N. 2001b.** Sex reversal and aromatase in chicken. *Journal of Experimental Zoology* **290(7)**:727–740 DOI 10.1002/jez.1123.

**Van Handel E. 1965.** Microseparation of glycogen, sugars, and lipids. *Analytical Biochemistry* **11(2)**:266–271 DOI 10.1016/0003-2697(65)90014-X.

**Van Handel E. 1985a.** Rapid determination of glycogen and sugars in mosquitoes. *Journal of the American Mosquito Control Association* **1(3)**:299–301.

**Van Handel E. 1985b.** Rapid determination of total lipids in mosquitoes. *Journal of the American Mosquito Control Association* **1(3)**:302–304.

**Wollman AL, Hamilton HL. 1968.** Direct action upon avian target organs by the antiandrogen cyproterone acetate. *Anatomical Record* **161(1)**:99–104 DOI 10.1002/ar.1091610110.

**Wollman AL, Hamilton JB. 1967.** Inhibition by an anti-androgen of stimulation provided by four androgenic compounds. *Endocrinology* **81(6)**:1431–1434 DOI 10.1210/endo-81-6-1431.

**Yin P, Li YW, Chen QL, Liu ZH. 2017.** Diethylstilbestrol, flutamide and their combination impaired the spermatogenesis of male adult zebrafish through disrupting HPG axis, meiosis and apoptosis. *Aquatic Toxicology* **185**:129–137 DOI 10.1016/j.aquatox.2017.02.013.

**Yu H, Wen K, Zhou X, Zhang Y, Yan Z, Fu H, Zhu J, Zhu Y. 2020.** Role of unfolded protein response in genital malformation/damage of male mice induced by flutamide. *Human & Experimental Toxicology* **39(12)**:1690–1699 DOI 10.1177/0960327120937049.

**Yu HM, Zhou XQ, Zhang YJ, Wen KX, Yan ZL, Fu H, Zhu YF. 2021.** Flutamide induces uterus and ovary damage in the mouse via apoptosis and excessive autophagy of cells following triggering of the unfolded protein response. *Reproduction Fertility and Development* **33(7)**:466–475 DOI 10.1071/RD20287.

**Zhang JL, Zuo ZH, Chen YX, Zhao Y, Hu S, Wang CG. 2007.** Effect of tributyltin on the development of ovary in female cuvier (*Sebastiscus marmoratus*). *Aquatic Toxicology* **83(3)**:174–179 DOI 10.1016/j.aquatox.2007.03.018.