# Peer review of "No effects of the antiandrogens cyproterone acetate (CPA), flutamide and p,p’-DDE on early sexual differentiation but CPA-induced retardation of embryonic development in the domestic fowl (Gallus gallus domesticus)"

_PeerJ, doi:10.7717/peerj.16249_

## Round 0.1 · original submission · Major Revisions

Dear authors,
Three reviewers have evaluated the manuscript and provided suggestions for improving the clarity of the manuscript. As you can see, the reviewers have pointed out the need to improve the explanation of the rationale of the study, methodological approaches and discuss the results in light of earlier published literature. Please indicate in your response letter how each of the reviewer comment was incorporated in the manuscript text and if it was not, please explain the reasons in the response letter.

Reviewer 1 ·

Basic reporting

I have revised the manuscript entitled “No effects of the antiandrogens cyproterone acetate (CPA), flutamide and p,p´-DDE on early sexual differentiation but CPA-induced retardation of embryonic development in the domestic fowl (Gallus gallus domesticus)”. This is an interesting paper showing that the embryo of G. gallus domesticus may be a suitable test system for the identification of substance-related mortality and developmental delays. The manuscript contains novel data that will contribute to the field of reproductive biology. However, some questions need to be answered.

1. Please add research hypothesis into the Introduction section, and improve aim of the study.

2. Is there any literature data showing the expression of androgen receptors in the embryo of G. gallus domesticus? In which structures? In which period of development? This information should be added into Introduction and discussed thoroughly.

3. It is difficult to follow results section and figures, i.e. there is a reference to Fig. 3A in the first paragraph of Results section, then 1A, then 2A – Figures and results description should be arranged sequentially.

Experimental design

The most important issue is p,p’-DDE. According to literature data (Hoffmann and Kloas 2016, doi: 10.1016/j.physbeh.2016.09.012; Muñoz-de-Toro et al. 2006, https://doi.org/10.1186/bcr1534) this compound may exert also estrogenic activity. Thus the rationale for using p,p’-DDE as antiandrogenic model compounds should be provided and discussed. Moreover, the obtained results refer to p,p’-DDE may derived from its estrogenic activity – it also should be discussed thoroughly.

Additional comments that should be addressed are defined as follows:
1. The scientific rationale for the CPA, flutamide and p,p´-DDE doses chosen need to be stated

2. Why day 19 has been examined?

3 . Please provide the detailed description of qPCR method.

Validity of the findings

The estrogenic action of p, p'-DDE should be discussed.

Additional comments

Overall, the subject of the study is of interest and in need of research. The manuscript is clearly written in professional language.

Reviewer 2 ·

Basic reporting

In this section, the authors conclude that compounds such as anti-androgens did not have an effect on gonadal sex differentiation in the chicken embryo, but some of the compounds were chick embryo toxic to development. This study is interesting, but in the introduction there is a lack of description of the significance of studying these three compounds, e.g. whether there are recent reports on the effects of these three compounds in sex differentiation. From the full text, there is a lack of recent reports in this area.

Experimental design

In "3.1 Embryonic mortality and malformations", the authors mention and describe a small number of malformation phenotypes at several concentrations, even in solvent controls, but without significant differences. These conclusions are not presented in tables or pictures, and the authors do not make clear the need to analyze them.

Validity of the findings

no comment

Additional comments

Line54-67: Same problem as elsewhere, not many references cited in the past five years。
Line104: The purity of other compounds also needs to be provided
Line 169: Pictures should be discussed in the text by serial number
Figure 3A:The statistical meaning of the labeled abc in all bar graphs should be further clarified
Figure3A: Why does morality not have an error line
Line205: P>0.001?
Line298-302:Tenses are mixed in the discussion section
Lin302-304: Language is not brief enough

Reviewer 3 ·

Basic reporting

1) The authors provide a characterization of the effect of three androgen antagonists, CPA, flutamide and p,p’-DDE in chicken development and gonadal differentiation. Despite the article is well written, the manuscript requireas a reorganization of either the figures or the text in the result section, as the first figure described in the manuscript is Figure 3. Moreover, there is a back and forth in the results section between the sections. I suggest that the authors either describe the results by treatment (keeping the figures as presented) or by technique/measurement (reorganizing the figures but keeping the text as it is).
2) Line 58, replace gender with sex, as gender is a social construct only present in humans and not to birds.
3) The introduction ends abruptly and should include at the end at least a paragraph summarizing the aim of the study and the reasoning behind it.

Experimental design

1) More detailed information is required for the methodology section. A detailed description of the methodology for extracting and measuring the lipids, glycogen and protein levels is required, as the papers cited are not accessible online.

Validity of the findings

1) Despite there is a lengthy description of the malformations obtained in the different treatment groups, they are not consistent and only happened in one or two embryos. This suggest that the drug treatments are not responsible of the phenotype obtained and it might be an intrinsic difference of individual treated or due to the vehicle (DMSO), as it was also shown to have similar phenotypic effects. This is also supported by the non-statistically significant difference between the treatments and the solvent or control groups. I suggest removing that section of the results.
2) The same applies to the mortality differences. It is mentioned that there is a concentration-dependent increase in the embryonic mortality, but only is significantly different in a one or two concentrations, instead in all of them.
3) Despite including the numerical data, the manuscript will be enriched by including representative data of the measured features. For example, showing the differences in skull, tarsometatarsus and ulna size between embryos with or without CPA treatment or H&E of the ovarian and testicular sections used for the quantification, to evaluate if the morphology is not affected despite of the size not changing.
4) A justification of why to the injections were performed at day 1 and samples collected a day 17 should be provided, as the sexual determination of the gonad occurs around day 6.5 and the first morphological differences can be detected at E8.5. One of the possible reasons of why there was not any phenotype in gonadal differentiation could be attributed to the half-life of the anti-androgenic compounds. Both flutamide and CPA have shorter half-life compared with the DDE. So, they might have been degraded by the time when gonadal determination occurred. This would be worth mentioning in the discussion, as knocking down of androgen receptor resulted in both lacunae and cortical cord defects in the ovary (Tanaka et al. doi.org/10.1016/j.mce.2017.01.008).
5) In addition, to study the effect of androgen antagonists, other organs that are known to be responsive to androgens could be better reporters, such as muscle, pituitary, or brain. This could also be expanded in the discussion.
6) Lastly, in the discussion it is mentioned that androgens appear to play a minor role in avian sex differentiation. That it is not totally correct, as the ovary requires androgens to produce estrogens and even contains more androgen producing cells than the testis. This is important as you need estrogens to induce the ovarian differentiation and cortex formation. So, androgens are required but androgen signaling might not be required, at least, for male gonadal differentiation.

---

## Round 0.2 · accepted · Accept

Thank you for addressing the reviewers' comments and modifying the manuscript accordingly. The manuscript is now ready for publication.

Reviewer 1 ·

Basic reporting

The manuscript was properly revised and I recommend it for publication.

Experimental design

no comment

Validity of the findings

no comment

Additional comments

no comment

Reviewer 2 ·

Basic reporting

The author has made corrections as requested and I have no additional comments.

Experimental design

The author has made corrections as requested and I have no additional comments.

Validity of the findings

The author made the correction as requested, and I have no additional comments.